# Multi-Agent Reinforcement Learning-Based Decentralized Controller for Battery Modular Multilevel Inverter Systems

**Ali Mashayekh** [1,*] , **Sebastian Pohlmann** [1] , **Julian Estaller** [1] , **Manuel Kuder** [2] , **Anton Lesnicar** [1] , **Richard Eckerle** [1] **and Thomas Weyh** [1]

1    Department of Electric Power Supply, Universität der Bundeswehr München, 85577 Neubiberg, Germany
2    Bavertis GmbH, 81929 Munich, Germany
*    Correspondence: ali.mashayekh@unibw.de

**Abstract:** The battery-based multilevel inverter has grown in popularity due to its ability to boost a system's safety while increasing the effective battery life. Nevertheless, the system's high degree of freedom, induced by a large number of switches, provides difficulties. In the past, central computation systems that needed extensive communication between the master and the slave module on each cell were presented as a solution for running such a system. However, because of the enormous number of slaves, the bus system created a bottleneck during operation. As an alternative to conventional multilevel inverter systems, which rely on a master–slave architecture for communication, decentralized controllers represent a feasible solution for communication capacity constraints. These controllers operate autonomously, depending on local measurements and decision-making. With this approach, it is possible to reduce the load on the bus system by approximately 90 percent and to enable a balanced state of charge throughout the system with an absolute maximum standard deviation of $1.1 \times 10^{-5}$. This strategy results in a more reliable and versatile multilevel inverter system, while the load on the bus system is reduced and more precise switching instructions are enabled.

**Keywords:** battery; battery management system; decentralized controlling; multilevel inverter; SoC balancing; multi-agent reinforcement learning; game theory; reconfigurable battery system

## 1. Introduction

Climate change has already emerged as one of the most pressing issues, and its impact will grow dramatically in the coming years [1]. As a result, reducing $CO_2$ emissions, particularly in the transport sector, is critical to meet global commitments [2]. In addition to the representative study [3], countless studies claim that electrifying automotive drivetrains is essential to reduce $CO_2$ emissions from the automotive industry. Conventional Electric Vehicles (EVs) continue to face market competition from internal combustion engine vehicles due to their limited range, higher acquisition cost, and long charging times. However, due to technological leaps and government policy requirements, a dominant role of lithium-ion batteries (LIBs) in automotive applications has been secured over the next decade, because of their high energy density and their long lifetime in combination with reduced prices [4,5].

A multilevel inverter (MLI) system, as previously established [6–10], enables an increased EV's propulsion efficiency in comparison to a typical two-level inverter [10,11]. Furthermore, by merging many independent tasks, such as the Battery Management System (BMS), the motor controller, and the charging system [12], the costs for an EV's drivetrain can be significantly reduced [13]. Aside from extending the driving range via increased inverter, battery, and motor efficiency, the monitoring capabilities of an MLI system for individual battery cells are widely improved compared to a conventional system. This enables optimal operation and distribution of stress on single cells, as well as the possibility to avert a whole system failure. Additionally, the system is beneficial in terms of a lower voltage level per module in comparison to a conventional battery system. However, as

mentioned by [8], research on MLI systems comprises an increasing number of modules to be integrated to form a single strand. Because of the rising complexity, this development impedes the communication of the system. As additional modules must be handled individually, more data must be delivered at each switching request [13]. Furthermore, because a sine voltage is synthesized more precisely with a higher number of voltage steps, in total, more switching commands are necessary [14]. Decentralized controllers emerged as a practical solution for communication capacity problems in a variety of applications [15–17] and, particularly, in MLI systems [13,18,19]. Each MLI module has a micro-controller that is utilized for the operation itself, rather than converting the data received by the master controller into a switching signal [20,21]. The accepted average discharge/charge rate of each module during a certain period of time are the only data that have to be transmitted from the master controller to all modules. The amount of data transmitted to each module is identical. This article describes the operation principle of an EV's battery pack, which is based on an MLI and operated by a decentralized controller. A simulation model of the system is developed in Matlab/Simulink to validate the concept.

## 2. Materials and Methods

Decentralized controllers have already proven to be an effective alternative for governing complex power systems [22,23]. They are based on the premise that numerous controllers can work independently, based on local measurements. Among all decentralization methods, Reinforcement Learning (RL)-based algorithms show promising results, including their dynamic response to new events, self-adaptation, and optimization, which eliminate the need for the time-consuming process of pre-tuning the key parameters. By interacting with the environment, an RL agent is modeled to make sequential decisions [24,25]. Typically, an infinite-horizon discounted Markov Decision Process (MDP) is used to model the environment. MDP has been widely used as a standard model to describe an agent's decision-making process with complete system state observability [26]. A branch of RL is called Multi-Agent Reinforcement Learning (MARL). The majority of effective applications include the interaction of several agents or players, which should be systematically represented as MARL issues [27]. The sequential decision-making issue of numerous autonomous agents, operating in a shared environment, is specifically addressed by MARL [28]. Each agent seeks to maximize its own long-term return by interacting with the environment and other agents. Also, it focuses on examining how different learning agents interact with one another in a common setting. Each agent acts to enlarge its own interests and is driven by its own rewards. In some contexts, these goals conflict with those of other agents, which leads to complicated group dynamics. Multi-agent systems, particularly repeated games and game theory, are all strongly connected to MARL.

The proposed decentralized control approach is based on MARL and game theory. It should be noted that game theory is the study of mathematical models describing the strategic interactions of rational agents. The exact definition of game theory varies based on the sort of game being played. The three most frequent game concepts are cooperative, competitive, and evolutionary games. For example, evolutionary game theory is the study of players, who adapt their strategies over time according to rules which are not necessarily rational or foreseeable [29–31]. Generally, each game consists of four steps, as shown in Figure 1.

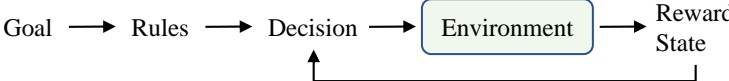

**Figure 1.** General four-steps methodology for decision-making of a reinforcement learning algorithm influenced by its environment.

### 2.1. Intelligent Switching System

While describing the MLI system as a MARL problem, the proposed approach, called Intelligent Switching System (ISS), attempts to find an appropriate solution to decrease

high communication efforts in the MLI system, while also meeting the two most important system goals, namely, State of Charge (SoC) balancing and correct modulation of the desired output signal. The first and most important stage in defining a MARL problem is to identify the relevant participants and the environment in which they are able to interact.

There are two different categories of players in this particular MARL problem. The controller for the master module, simply called "master", and the controller for each slave, simply called "cell". The master interacts in "games" with several groups (packs), each made up of twelve cells. Figure 2 illustrates the submodule environment with the cell structure.

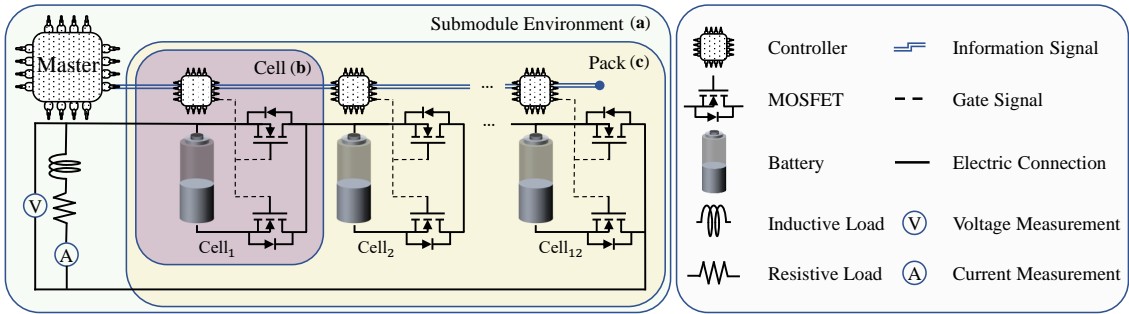

**Figure 2.** Simulation structure of the used battery system's pack. (**a**) Interaction environment between the master and a pack. (**b**) Cell structure based on two semiconductor switches' topology. (**c**) Interaction environment between the cells.

As a result, two environments can be defined. First, the interaction environment between the cells, known as submodules. In the submodule environment, participating cells actively strive to achieve a balanced SoC within the designated submodule. They rely on passive information received from the master to make decisions and take appropriate actions. Second, the environment between the submodules and the master is referred to as the module. The master actively works in the module environment to balance the SoC between packs. This is accomplished by continuously allocating the required charge among the participating packs based on their previous states and providing essential and relevant information to each submodule environment. Figure 3 illustrates the module environment structure.

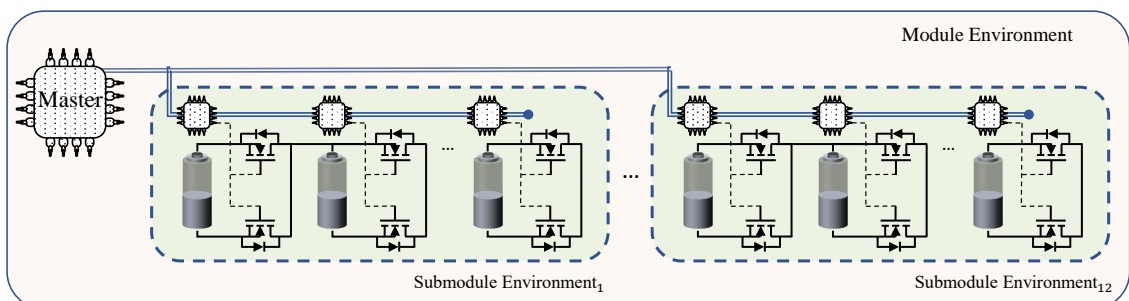

**Figure 3.** Simulation structure of the used battery system's pack to generate the AC waveform voltage. The overall module environment is based on several submodule environments.

The initial step in developing the ISS involves identifying the individual goals of every player (following the establishment of the participants and the interaction environment). The master is responsible for ensuring that the desired output signal has the necessary modulation. Cells, on the other hand, are focused on achieving the specified switching strategy, which may include objectives such as SoC, temperature, or State of Health (SoH) balancing.

The next stage is to establish the communication protocols that govern the interactions among the various components. These details determine the methods by which data are shared between the players, the master, and the individual cells or packs.

During the controlling of a conventional MLI system, the master has access to the output voltage and current in each cell and has to supply exact information to each one in order to achieve the desired switching behavior at a specific time. This requires the master to process and communicate a large amount of information in real time, which can place a significant strain on the communication bus and slow down the system's response time. To overcome this limitation, the proposed MARL system utilizes a different approach. In this system, the master has access to the output voltage and current of each pack, rather than the information of a single cell. In other words, the master can process and broadcast information at a higher level, which is more efficient and less burdensome for the communication bus. Specifically, the master broadcasts two pieces of processed information to each cell in the pack, which are called "Reference" and "Ticket". These terms are explained in more detail later in the system's design.

As mentioned before, the design of the MARL system allows only two distinct non-bidirectional signals to be used for communication between the master and the cells. The master has two main functions in the designed MARL system. Firstly, it attempts to generate the required number of in-series switched cells in each pack to achieve the desired output voltage using the nearest level modulation approach. This approach involves adjusting the sum of the output voltage of the participant's cells to the nearest possible voltage level, in order to minimize the voltage error and ensure accurate voltage modulation [32,33]. Secondly, the master computes the permissible Average Depletion Charge (ADC) per pack regarding the Pack Size (PS), which the cells utilize as one of their decision-making reference values. The ADC is shown in Equation (1). The cells continually examine the reference signal in order to have a better understanding of the interacting environment.

$$\text{ADC} = \int \frac{I_{\text{Pack}} dt}{\text{PS}} \tag{1}$$

The decision-making process of the participants is the most important and complex part of the MARL system. The master generates the ticket signal to flag its decision. The ticket signal is broadcast to all cells in the system, and each cell must make a vital decision on its switching state based on this signal. This decision is based on the individual measurements of the cells as well as the received reference and ticket values, allowing each cell to synchronize its switching status with the master's decision. Using these values, each cell attempts to make an informed judgment regarding its switching state, ensuring that the output voltage is modulated correctly, while it aims to keep its own calculated ADC aligned with the permitted ADC per cell (reference signal).

As a rational player in the game, each cell seeks to maximize the likelihood to reach its objective. The objective of each cell can be formulated as a Mixed Integer Linear Optimization (MILO). The goal of MILO is to identify the optimal decision variable values that minimize or maximize the objective function while satisfying a set of linear constraints. The decision variables can be a combination of continuous and discrete variables, where the discrete variables are typically limited to integer values [34,35]. Each cell in the system considers the calculated ADC values as a continuous variable. Furthermore, the cells can operate in one of two modes: series or bypass, which is represented as a binary variable. In addition, the objective function was selected as a Cartesian product to enable a full examination of the optimization issue. Additionally, it allows the objective function to represent the complex relationships and interdependence among the variables involved.

The optimization approach is designed to consider several constraints. First, it provides the maximum absolute difference permitted between the estimated ADC and the allowable ADC (referred to as TolQ). It also takes into account the maximum absolute difference between the estimated Open Circuit Voltage (OCV) and the measured terminal voltage (TolU). The second constraint that is taken into account is the consecutive participation of instances in order to reach the output voltage, which is known as the Continuous Series Limit (CSL). Finally, the framework includes the number of successive occurrences in which the computed ADC is either higher or lower than the permissible ADC over a

certain period of time. This is referred to as the Sign Signal Limit (SSL). The values of the defined boundary conditions are given in Table 1.

**Table 1.** Boundary conditions for the cell's optimization approach.

| Boundary Condition | Value | Unit |
|:---:|:---:|:---:|
| TolQ | 0.006 | Ah |
| TolU | 0.01 | V |
| CSL | 10 | - |
| SSL | 25 | - |

However, since cells have limited knowledge of the strategies of other players, every decision is made without complete information, which introduces an element of uncertainty to each decision. Despite this uncertainty, each cell will try to determine the probability of success associated with various decisions and then choose the one that appears most likely to lead to a favorable outcome. Even though each decision is made without adequate and comprehensive knowledge, the cells evaluate their choices at the next time step and the decision-making process is rewarded/punished based on the outcome. This feedback loop can be used to refine the strategies of each player over time, as the cells learn from their successes and failures. Overall, there is no definitive right or wrong choice; the cells will continuously adjust their strategies based on their observations of the available evidence and the outcomes of their decisions as a feedback loop.

The decision-making process is divided into three stages. The first step is to create a cost function or to translate evidence into knowledge or tendency. To summarize this process, four types of evidence are employed. The first category involves the absolute error ($f_e$), which is the difference between the computed ADC of each cell (calculated separately) and the master-determined permitted ADC, which is the reference signal. The second category contains the actual physical position of a cell with respect to the adjacent cells ($f_{pp}$). The difference between the actual output voltage and the OCV of each cell is included in the third group as the OCV error ($f_{OCV}$). The last group incorporates the self-evaluation error ($f_{SE}$) or a corresponding reward/punishment, depending on the consequences of previous actions. The tendency is summarized in Equation (2), derived from observed evidence and the effectiveness matrix A, which determines the desire of a specific cell towards serial switching. Matrix A is a collection of optimized constant values that define the effectiveness of each individual piece of evidence. The values of the effectiveness matrix A are given in Table 2.

$$\text{Tendency} = \begin{bmatrix} a_1 & a_2 & a_3 & a_4 \end{bmatrix} \times \begin{bmatrix} f_e(Q_{\text{self}}, Q_{\text{Ref}}) \\ f_{\text{pp}}(\text{PP}) \\ f_{\text{ocv}}(V_{\text{ocv}}, V_{\text{self}}) \\ f_{\text{SE}}(\text{feedback loop}) \end{bmatrix} \quad (2)$$

**Table 2.** Matrix A index values.

| Matrix A Index | Value | Factor |
|:---:|:---:|:---:|
| $a_1$ | 0.8 | $f_e$ |
| $a_2$ | 0.5 | $f_{pp}$ |
| $a_3$ | 0.4 | $f_{ocv}$ |
| $a_4$ | 1.0 | $f_{SE}$ |

It should be emphasized that the goal of the cost function is not to give all players the same tendency, but rather to offer them an equal chance to reach the same conclusion. In other words, Switching Chance (SC) expresses the likelihood that the serial switching state is correct. Because each cell must make a choice in turn due to its physical positioning, the probability that two cells will make the same conclusion drops rapidly from prior

to subsequent cells if the decisions are made based on equal tendency. This is shown in Figure 4, where switching chance and tendency are plotted over the physical position.

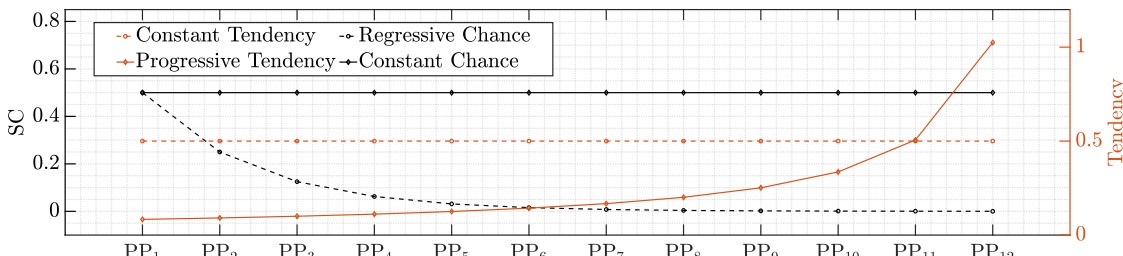

**Figure 4.** Comparison between constant and progressive tendency, and their effect on desirable SC. Tendency and its corresponding SC are plotted over the physical position (PP) of the cells.

In the second step of the decision-making process, the estimated tendency based on the cost function is applied to the option's spectrum, which is presented with the probability of each decision. The goal of this step is to use probability theory to determine a boundary between two potential switching alternatives in the option's spectrum.

In the given system, each cell is equipped with a two-semiconductor switch topology, enabling them to switch between two distinct states: bypass and series, which are depicted in Figure 5a,b, respectively. The "series" state implies that the cell is added to the output voltage of the entire pack (the cell is depicted in red), while the "bypass" state refers to the cell being excluded from the output voltage (the cell is depicted in gray). In other words, an individual cell can achieve two distinct output voltage levels according to Equation (3), with $V_t$ being the terminal voltage of the battery cell. Furthermore, the total output voltage ($V_{out}$) of a single-phase strand with $n$ cells can be described using Equation (4), where $V_{out,j}$ is the output voltage level of each individual cell [14].

$$V_{out,n} = \{1; 0\} \cdot V_{t,n} \tag{3}$$

$$V_{out} = \sum_{j=1}^{n} V_{out,j} \tag{4}$$

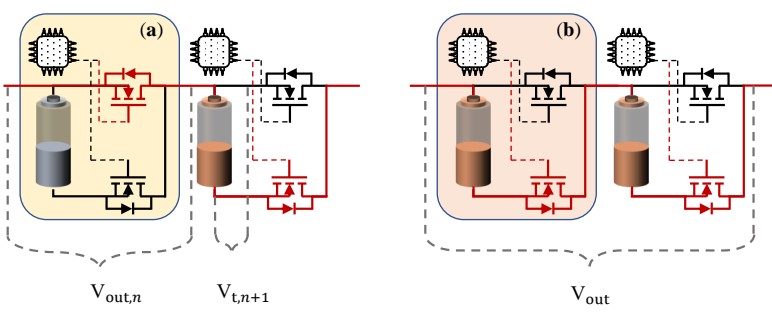

**Figure 5.** Comparison between (**a**) bypass and (**b**) serial switching state.

It is expected that the likelihood of the pack being in either one of these two states is equivalent to one, as these two states comprise the only possible options. A linear function (see Equation (5)) was employed to scale the computed tendency from zero to one and transfer it to the SC. This ensures that the decision-making process remains intuitive. Furthermore, the bias value was incorporated into the decision-making process to add randomness and mutation. Regardless of a player's physical position or previous behavior, they have the same initial minimum chance of making the same decision. This prevents the decision-making process from becoming overly predictable or repetitive, which could avoid converging at local minima. By incorporating instinct and mutation into the decision-

making process, it becomes more adaptive and capable of dealing with a broader range of events and factors.

$$SC = (1 - \text{bias}) \times \text{Tendency} + \text{bias} \tag{5}$$

The final decision is made by generating a random number from a uniform distribution. As noted before, there is no single decision that is definitely right or wrong. On the contrary, choices are made based on their probability of being correct. By adapting to the environment through time, even options with a lower probability of success may still be a viable decision for the system. This approach can be seen as a modified response to a given event, as it results in a more nuanced and optimized approach to both the generalization and the cost function. By introducing randomness into the decision-making process, the system becomes better equipped to handle a range of possible outcomes and scenarios. This can lead to a better overall performance and a more robust decision-making process, as the system is not overly reliant on any single decision or strategy.

Once a decision has been made, the next step is to update the ticket signal accordingly. For example, if the decision is to change the cell's status from bypass to serial, then the ticket signal should be decreased by one and passed on to the adjacent cell. This process of lowering the ticket signal refers to the action of occupying the necessary series switched option by the corresponding cell. It is important to note that the decision-making phase can only begin if the ticket signal is greater than zero but less than the actual location of the cell. If the ticket signal falls outside this range, then the decision is bypassed and the cell's status remains unchanged or is converted to series, depending on the situation. If the ticket signal value exceeds the physical position of the relevant cell, it signifies a system problem or this value returns a special command to the cells, as the requested number of in-series switched cells exceeds the available number of cells. If the ticket signal is equal to zero, this can be interpreted as all remaining cells in the sequence needing to be switched to the bypass state. On the other hand, if the ticket signal is equal to the physical position of the cell, all cells must be switched to series, and no further processing is required.

The "ticket signal" has a dual purpose: it is not only used to communicate the precise number of switched cells in a battery pack, but it can also convey specific commands with specific values. For example, to minimize switching cases and losses, as well as to reduce computational effort, switching is intended to occur only once at each voltage level. As a result, the master always verifies whether the number of designated switching cases for a given submodule is the same as the previous command. If the required number of in-series switched cells is the same as the previous command, the master controller changes the ticket value to 100, indicating that each cell should maintain its prior switching state.

### 2.2. Self-Evaluation

In the last step, each agent (cell) evaluates itself based on its interactions with the environment and the effect of its previous decisions is recorded. The assessment is then utilized to optimize the cost function by modifying the self-evaluation function ($f_{\text{SE}}$).

In essence, the agents' behaviors and responses to their environment are constantly modified via a feedback loop that considers the knowledge gathered by each agent during past interactions. This feedback loop contributes to the overall optimization of the system performance by allowing agents to change their behavior based on current conditions and previous experiences. There are two types of post-action self-evaluation, which are a fixed and a floating reward/punishment. The fixed reward/punishment is designed to compensate for and avoid large errors that cause the tendency value to shift dramatically. Thus attempts to influence the agent's persistent behavior by directly impacting the tendency to result in the opposite direction of the insistent action. The aim of this approach is to keep the agent's performance within an acceptable range, even when the environment is changing and unpredictable.

The floating reward/punishment method achieves this by setting a target performance range and adjusting the reward/punishment signal depending on the agent's current performance relative to that range. For instance, if the agent's performance is within the

acceptable range but has a diverging gradient from the goal range as compared to the previous time step, a floating reward/punishment is created to address this divergence gradually. However, as mentioned in Section 2.1, the agent may continue to make the same decision despite changes in its tendency, displaying persistent behavior. When an agent performs persistent behavior or violates the allowed goal range, a fixed reward/punishment is created to shift the agent's tendency towards the minimum or maximum, with the objective of changing its behavior in the opposite direction of the violation.

By adapting the reward/punishment signal in response to changes in the agent's performance and the environment, the floating reward/punishment method can help to maintain high performance, even in dynamic and unpredictable situations. However, it is worth noting that this approach can be more complex to implement and may require more computational resources than fixed reward/punishment methods.

The last cell in each submodule faces unique challenges. While the cost function aims to ensure that all cells have equal decision-making opportunities, the last cell may not have the same options as the other cells. The last cell may need to compensate for the possible errors of the other cells, resulting in a switching state, regardless of its own preferences and self-evaluation. To reduce this effect, it is crucial to prioritize the performance of cells located earlier in the module. This can be accomplished by applying a higher punishment rate than rewards. The cell's decision-making process is summarized in Figure 6.

To avoid unforeseen events and maintain control over the output voltage, the master utilizes a feedback loop alongside the cells, which also helps to update its knowledge about the dynamic output voltage of the cells. Even though the master cannot access the output voltage of each individual cell, it plays an important role while controlling the output voltage. Through the feedback loop, the master agent estimates this information by dividing the output voltage of the entire pack by the number of cells participating (in this case, 12) and updates its estimation in each iteration. However, due to the constantly changing behavior of individual cells and the varying SoC and SoH conditions within a pack, it is impossible to estimate the output voltage of each cell with perfect accuracy. Nevertheless, the level of precision achieved is sufficient for effective modulation. The master's decision-making process is summarized in the flowchart in Figure 7.

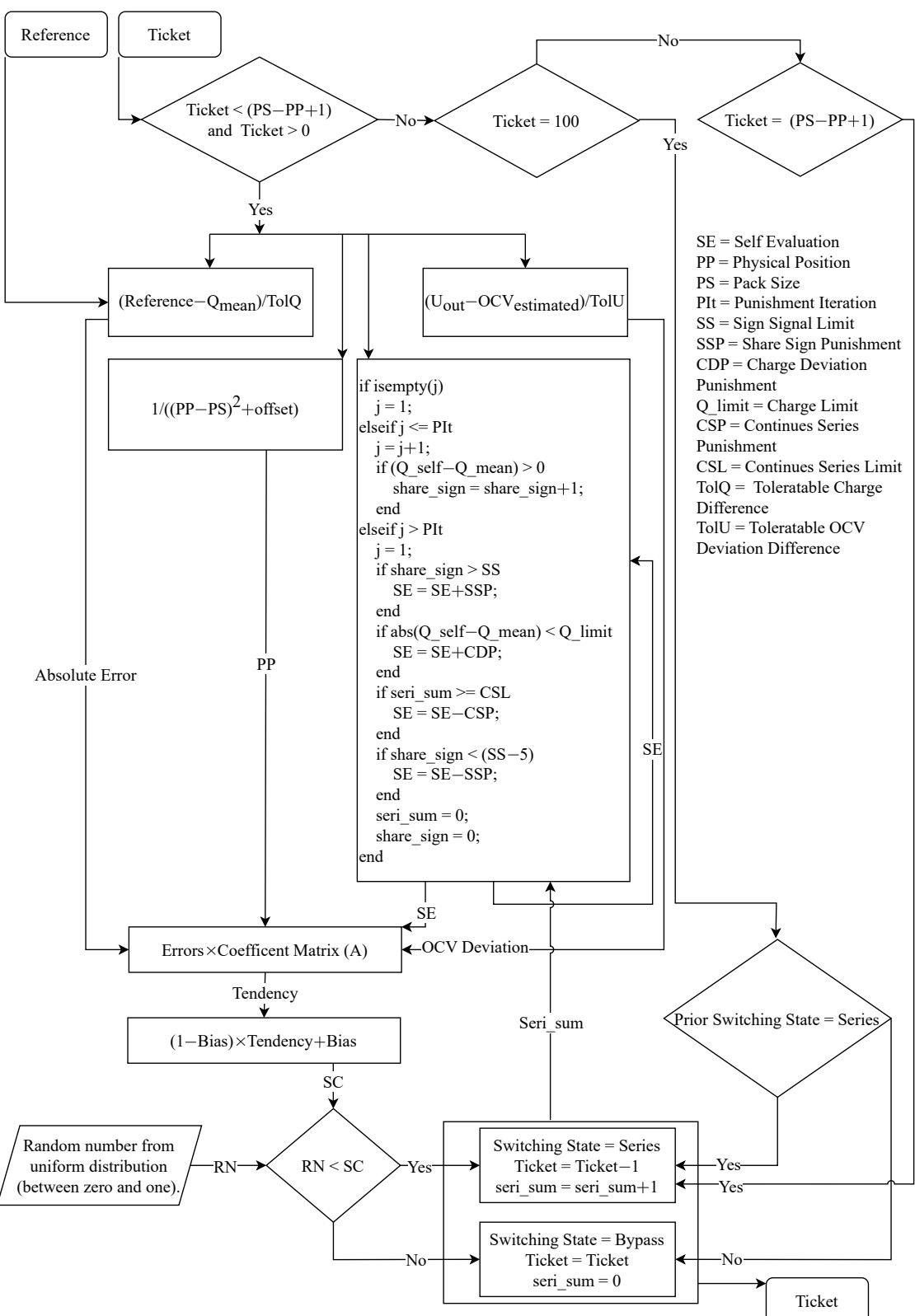

**Figure 6.** Summary flowchart of the cell's decision-making process.

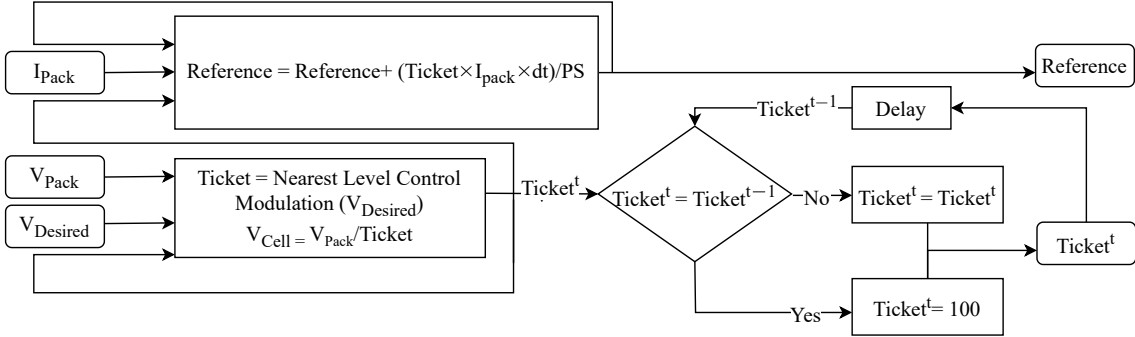

**Figure 7.** Summary flowchart of the master's decision-making process.

## 3. Simulation Structure and Results

This section outlines the procedure for simulating the MLI-based battery system with the applied decentralized algorithm. The simulation was set up using MATLAB/Simulink and involved establishing a single strand of the MLI system with two semiconductor switches implemented as ideal switches in each submodule, based on the Modular Multilevel Converter (MMC) topology [11]. A LIB cell with a voltage source dependent on the SoC was modeled using a look-up table with the OCV. The SoC of individual battery cells was measured using Coulomb counting due to the level of abstraction regarding the utilized battery cell. To accurately model the battery's behavior with respect to the actual cell-current, dynamic voltage variations were integrated using a first-order equivalent circuit model with one RC module. The battery capacity and dynamic parameters were established based on testing and modeling the 18650 battery cell LG HG2, to ensure the simulations were as precise as possible and the balancing procedure could be verified. Additionally, a serial resistive and inductive load were included in the current path to simulate an electric motor controlled by the MLI system. The resistive and inductive impedance could be adjusted according to the requirements.

### 3.1. Submodule Environment

In the first step, the aim was to maintain a low level of complexity in the simulation by initially limiting it to one submodule or interaction environment between twelve cells and a master. Further discussion on the other interaction environment 'module' is in Section 3.2. The boundary conditions and hyperparameters are given in Tables 3 and 4.

**Table 3.** Boundary conditions for the simulation's parameters.

| Boundary Condition | Value | Unit |
|:---:|:---:|:---:|
| Module count | 1 | - |
| Cells per module | 12 | - |
| Update frequency | 50 | kHz |
| Resistive load $R_{\text{load}}$ | 5 | Ω |
| Inductive load $L_{\text{load}}$ | 0.5 | mH |

The resulting modulated voltage is depicted in Figure 8a. As previously stated, the master attempts to keep the switching cases as low as possible in order to reduce switching losses. Thus, a new switching command is generated once at each voltage level, and, during the on-voltage level, the ticket signal remains equal to 100, indicating that the previous switching state is still in effect, as depicted in Figure 8b.

**Table 4.** Tuned hyperparameter values for the simulation.

| Hyperparameter | Value | Abbreviation |
| --- | --- | --- |
| Punishment iteration | 50 | PIt |
| Share sign punishment | 0.1 | SSP |
| Charge deviation punishment | 0.2 | CDP |
| Continues series punishment | 0.7 | CSP |

As an example of a cell's behavior, the output voltage and current response of the cell positioned on the fourth physical position are shown in Figure 9c,d, respectively. Additionally, its self-evaluation and the consequent tendency are depicted in Figure 9a,b, demonstrating the cell's high tendency (or comparatively higher likelihood of switching) in a series switching condition, followed by a lowered self-evaluation signal and a decreasing tendency value in the following time steps, and vice versa.

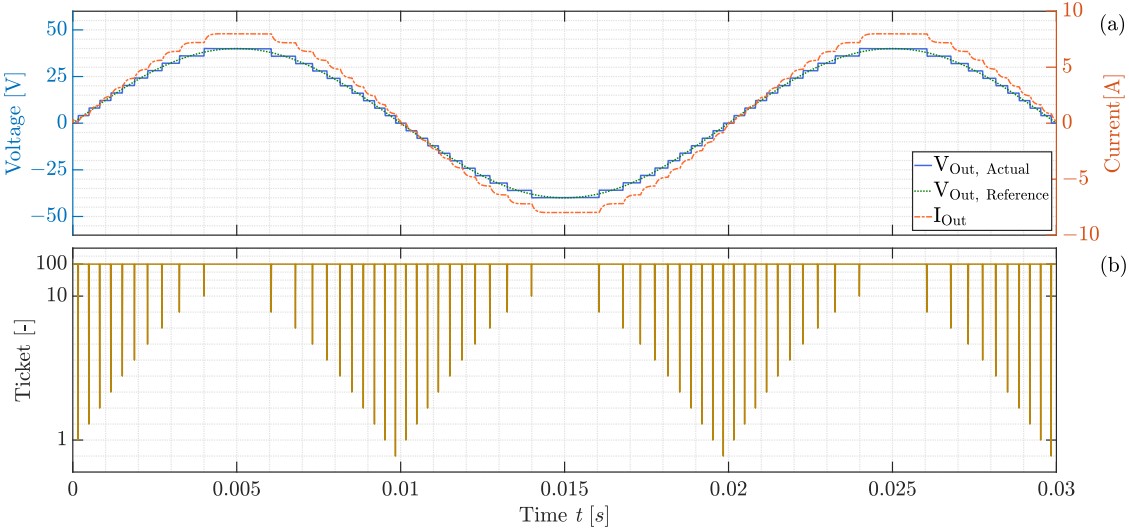

**Figure 8.** (**a**) Actual modulated voltage compared to the reference voltage; (**b**) corresponding ticket signal.

The standard deviation of the SoCs was used to evaluate the quality of the batteries' SoC balancing (see Equation (6)). The standard deviation is a measure of the variation in a set of values. A low standard deviation implies that the values in the set are frequently close to the mean, whereas a high standard deviation suggests that the values are more widely spread out throughout a greater range. As demonstrated in Figure 10a, the system begins entirely balanced, with each cell starting at 90% SoC. It remains balanced during discharging, with a maximum absolute standard deviation of $6 \times 10^{-6}$. Nevertheless, due to non-optimized hyperparameters—notably, the self-evaluation function '$f_{\text{SE}}$' of individual cells—the cells initially exhibit divergent behavior, leading to an increase in the absolute magnitude of the standard deviation among the cells. Following the initial feedback loop-based optimization phase, the agents' self-evaluation signal is constantly optimized with respect to the unexpected behaviors of other agents. This causes variations in the resulting standard deviation, which tends to converge around a specific value.

$$\sigma = \sqrt{\frac{1}{N} \sum_{i=1}^{N} (SoC_i - \mu_{\text{SoC}})^2} \tag{6}$$

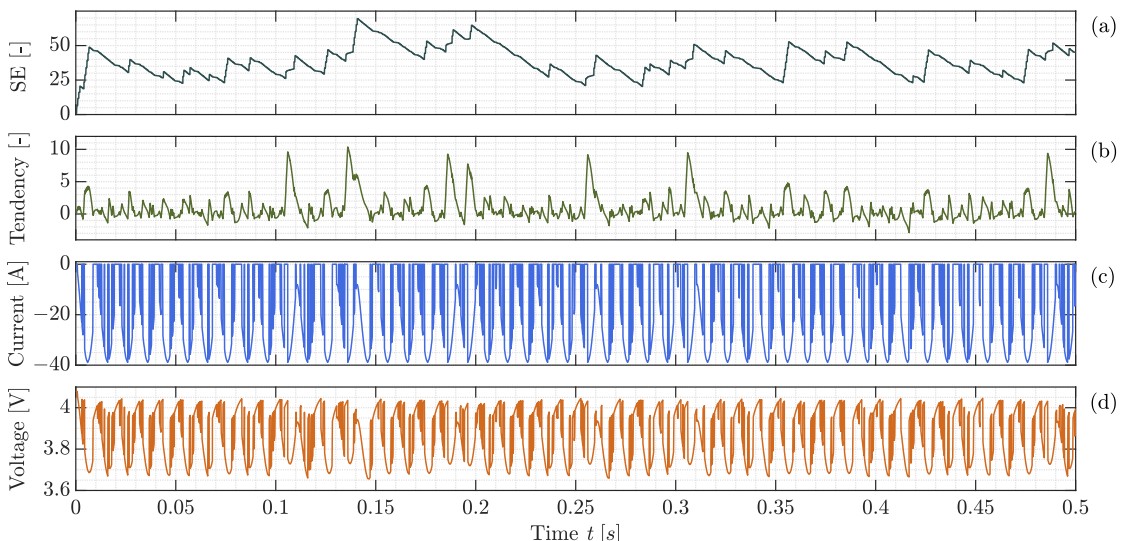

**Figure 9.** Examples of (**a**) calculated self-evaluation (SE) value; and resulting (**b**) tendency, (**c**) response current, and (**d**) voltage of a specific cell (cell number 4).

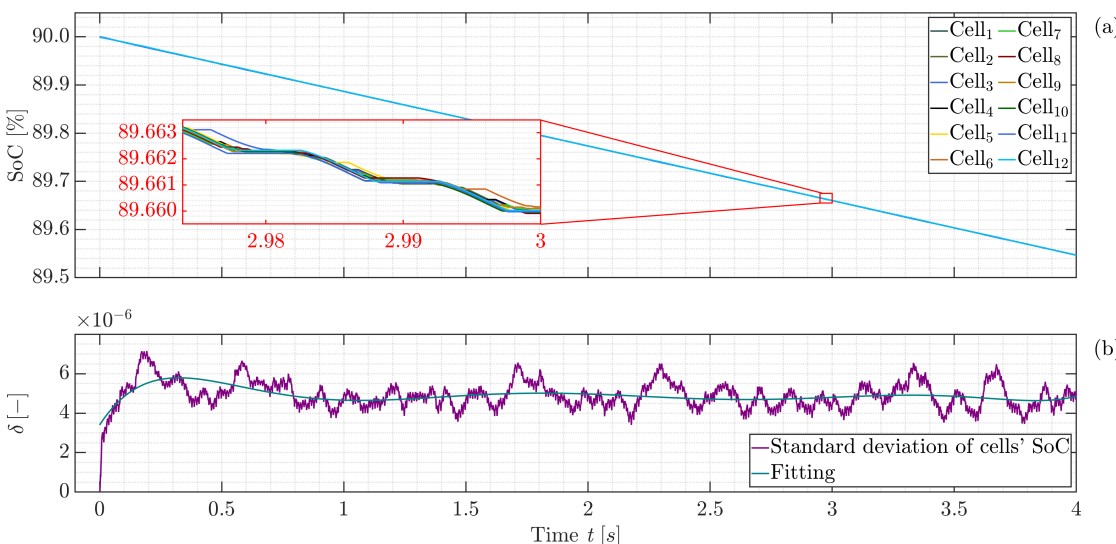

**Figure 10.** (**a**) SoC balancing result of 12 cells' interaction, and (**b**) the associated standard deviation.

### 3.2. Module Environment

It was demonstrated that the objectives of modulation and SoC balancing at a low-level environment, called the submodule, which involved the interaction between the master and a pack of 12 cells, could be controlled in a decentralized manner using MARL. The subsequent step was to concentrate on the high-level environment (module), which involved the interaction between the master and 12 submodules in a new practical operation scenario that included the modulation of a voltage waveform with an amplitude of 400 V. The game, or environment, consisted of one master and 12 packs, each with 12 cells. It should be noted that each pack generated its own environment, and the actions and reactions of each environment were distinct from those of other environments (packs). The boundary conditions and hyperparameters are given in Tables 4 and 5.

In the new game (environment), all agents (cells) were allowed to perform as they did in the previous game. Each cell was only able to engage in the low-level environment, and their goal and access to information remained unchanged. However, the master's new objective was to provide the ticket signal to each pack so that the output voltage was

accurately modulated and the SoC was balanced among the packs, by guaranteeing that the ADC was the same for all packs.

**Table 5.** Boundary conditions for the simulation's parameters.

| Boundary Condition | Value | Unit |
|---|---|---|
| Module count | 12 | - |
| Cells per module | 12 | - |
| Update frequency | 50 | kHz |
| Resistive load $R_{load}$ | 10 | $\Omega$ |
| Inductive load $L_{load}$ | 0.5 | mH |

It was ensured that each pack created a unique environment which was distinct from other packs. The actions and reactions that occurred within each environment were specific to that pack alone. Consequently, this enabled the distinction between the goals of the agents within each pack.

The cells in different packs may have had diverse objectives that were caused by a variety of factors. One of these factors was the performance properties of the cells. For example, high-power or high-energy cells may have been present in different packs, while another pack may even have had second-life cells that were optimized for a different purpose.

Another factor that may have contributed to the diverse objectives of cells was their physical positioning. For example, cells that were located in a specific part of the pack may have been more attuned to higher controlling changes in temperature. The focus on precise temperature observation could have led to a different set of goals for the cells in that specific pack compared to cells in another pack.

As demonstrated in Figure 11a, the master was capable of modulating the 400 V output voltage by dividing the required output voltage among 12 packs. Figure 11b depicts the unique ticket signal generated for each pack.

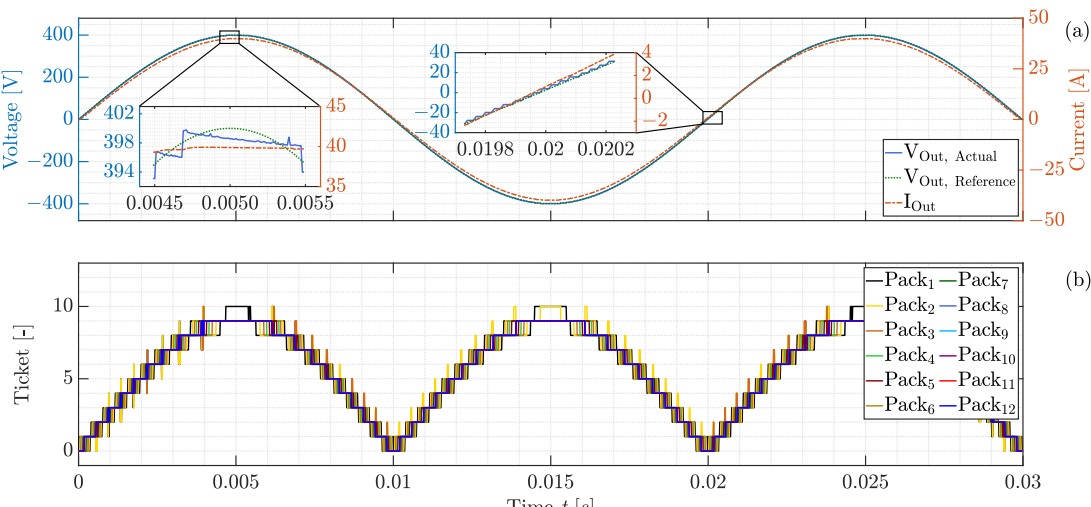

**Figure 11.** (**a**) Actual modulated voltage and resulting current compared to the reference voltage, and (**b**) the corresponding ticket signal to each pack.

In the new environment, the generation of the ticket signal was conducted based on two criteria. On the one hand, the master sent a special ticket signal to limit the switching losses across the cells and to favor maintaining the prior switching state. In the ongoing scenario, the master also sought to fulfill this purpose by allocating the same ticket signal to a given pack for as many time steps as feasible to decrease the number of switching occurrences at each cell. On the other hand, the simulation framework restricted the master's access to each pack's output voltage and current. With this limited information,

the master calculated the ADC of each pack during a set time period. According to this, the master ensured that the ADC remained out of balance across the packs by giving the matching number of ticket values to each pack. For example, the higher the ADC of a pack, the smaller the number of tickets assigned to it, and vice versa.

The Figure 12a depicts the ADC trend of the packs, demonstrating that the master maintained the balance among all packs by allocating adequate tickets to each pack. Furthermore, each pack was a separate and autonomous interaction environment capable of maintaining balance among its 12 constituent cells. As a result of all packs being balanced through the higher-level interaction between the master and the 12 packs while each lower-level environment (12 cells) was balanced independently, it can be deduced that all cells, as part of the operating system, could maintain the SoC balance throughout the system.

As previously noted, SoC balancing at the system level involves interaction between cells inside a pack as well as between packs, which can result in higher errors than balancing cells within a single pack, due to the accumulation of errors between these two levels. Figure 12b shows that the divergence of the SoC of cells at the system level increased significantly at first due to self-hyperparameter tuning, but gradually converged to a constant value of $1.1 \times 10^{-5}$. Furthermore, Figure 13 illustrates the estimated average cell's voltage.

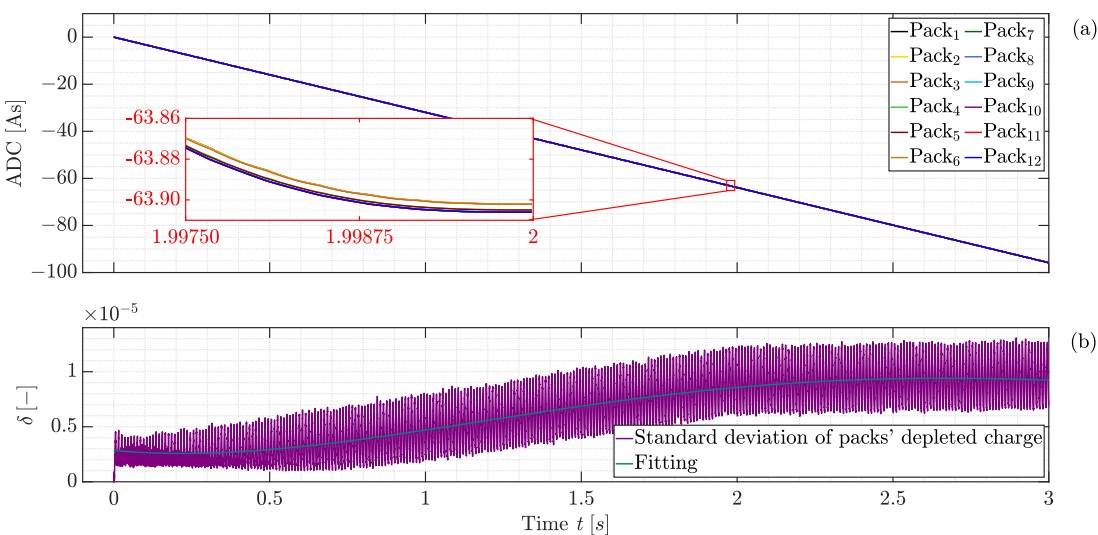

**Figure 12.** (**a**) ADC for each pack's course, and (**b**) participants cells' SoC standard deviation.

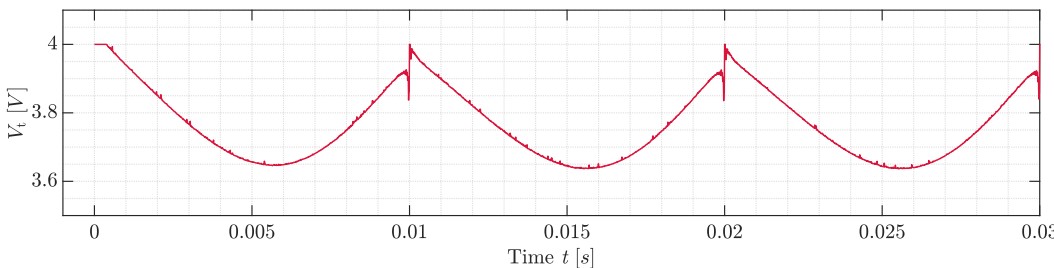

**Figure 13.** Estimated average cell's voltage.

### 3.3. Discussion

The system illustrated in Figure 12 provides a clear example of the potential applications of the offered decentralized controlling method. One conceivable application that immediately comes to mind is to use this to power an electric motor. The decentralized controller depicted in this figure plays a crucial role in ensuring that the system voltage is accurately established from the outset, which is crucial for achieving smooth operation, particularly when operating a motor.

Furthermore, starting with a well-balanced system, individual battery SoC values converge to an absolute maximum standard deviation of $1.1 \times 10^{-5}$ within the system. This suggests that battery cell balancing can be accomplished inherently using a decentralized controller, which is a substantial benefit over conventional systems that need external balancing circuits or additional hardware.

The decentralized controller also increases the system's resilience to unexpected events by allowing each cell to make autonomous decisions based on its own information, decreasing the impact of any individual cell errors. In the event of a failure, the feedback loops of other cells can be used to investigate the impact of an individual cell's behavior on the submodule's supplied output voltage, ensuring the system's continued stability and functionality. In other words, the cell's defective behavior is defined as the persistent decision to not contribute to the output voltage or to flip to the bypass state, pushing other cells to compensate for this behavior by deviating more from the reference signal.

## 4. Conclusions

Using a decentralized controlling technique based on MARL, this work proposes a unique way of regulating MLI-based battery systems. Unlike traditional MLI systems, which rely largely on a fast and reliable bus system, the suggested method greatly decreases the load on the master controller and minimizes the communication between the master and individual cells, which is traditionally a key bottleneck.

By reducing communication to just two broadcast signals from the master controller to all cells, the proposed method effectively decreases the bus load. This architectural modification enhances system efficiency and reduces the constraints associated with the number of modules in the traditional MLI setup. As a result, the system becomes more flexible and adaptable and is further capable of supporting a greater number of modules to meet the evolving demands of EVs.

Furthermore, the self-learning characteristics of the system enable it to adapt to changing situations and optimize its performance over time. Moreover, this offers advantages in terms of reduced complexity of finding hyperparameters and initial value sets, requiring less effort than conventional decentralized controlling systems. Additionally, because each submodule's internal interaction and decision-making process is independent, the proposed system allows separate submodules to pursue different objectives in addition to their contribution to the correct modulation of the desired voltage. This adaptability allows the system to handle a variety of goals while maintaining overall voltage management.

All in all, the proposed MARL-based decentralized controlling method presents a promising solution for regulating MLI-based battery systems in EVs. It offers advantages such as reduced communication overhead, enhanced flexibility and scalability, simplified configuration, and the ability to pursue diverse objectives.

In future research, it is planned to examine the application of different objectives across numerous submodules. All submodule environments in this article share a common goal of SoC balancing. Alternative objectives, such as temperature balancing and SoH balancing, can be pursued, taking into account the physical arrangement of each submodule and the aging characteristics of the battery modules. Furthermore, the pursuit of multi-target behaviors, such as simultaneous SoC and temperature balancing, could lead to promising results. In this environment, the creation and optimization of new goals becomes critical. In addition, by applying a fuzzy logic algorithm, multiple—and sometimes competing—goals can be integrated, allowing for a synergistic approach toward a better operation scenario.

**Author Contributions:** Conceptualization, A.M.; methodology, A.M.; software, A.M.; validation, A.M.; formal analysis, A.M.; investigation, A.M.; resources, A.M.; data curation, A.M.; writing—original draft preparation, A.M.; writing—review and editing, A.M., S.P., J.E., A.L. and M.K.; visualization, A.M.; supervision, M.K., R.E. and T.W.; project administration, M.K. and R.E.; funding acquisition, M.K., R.E. and T.W. All authors have read and agreed to the published version of the manuscript.

**Funding:** This research is funded by Munich Mobility Research Campus (MORE) as part of dtec.bw—Digitalization and Technology Research Center of the Bundeswehr ,which we gratefully acknowledge. dtec.bw is funded by the European Union—NextGenerationEU.

**Data Availability Statement:** Not applicable.

**Conflicts of Interest:** The authors declare no conflicts of interest.

## Abbreviations

The following abbreviations are used in this manuscript:

| | |
|---|---|
| ISS | Intelligent Switching System |
| EVs | Electric Vehicles |
| BMS | Battery Management System |
| LIBs | Lithium-ion Batteries |
| MLI | Multilevel Inverter |
| RL | Reinforcement Learning |
| MARL | Multi-Agent Reinforcement Learning |
| MMC | Modular Multi Level Converter |
| ADC | Average Depleted Charge |
| PS | Pack Size |
| MILO | Mixed Integer Linear Optimization |
| SC | Switching Chance |
| MDP | Markov Decision Process |
| SoC | State of Charge |
| SoH | State of Health |
| OCV | Open Circuit Voltage |
| GT | Game Theory |
| Ref | Reference |
| PP | Physical Position |
| PIt | Punishment Iteration |
| SSL | Sign Signal Limit |
| SSP | Share Sign Punishment |
| CDP | Charge Deviation Punishment |
| CSP | Continues Series Punishment |
| CSL | Continues Series Limit |
| RN | Random Number |

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
