# Peer review of "Multi-Agent Reinforcement Learning-Based Decentralized Controller for Battery Modular Multilevel Inverter Systems"

_electricity, doi:10.3390/electricity4030014_

Round 1

Reviewer 1 Report

The optimization problem and its formulation raise several questions that require further clarification:

  1. Could you please specify the type of the optimization problem addressed in your study? Additionally, it would be helpful to understand the constraints considered in the formulation.
  2. Can you provide information about the number of variables in the optimization problem? It would be valuable to know the distinction between free variables and state variables, if applicable.
  3. The objective function in your paper is formulated as a Cartesian product. Could you please explain the rationale behind this choice? Why were no other types of products considered for the objective function?
  4. The selection of the matrix A is not clearly explained. Could you provide more details on how this matrix is determined and its significance within the optimization problem?
  5. Additionally, has a sensitivity analysis of the four parameters (a1 to a4) mentioned in the paper been conducted? If so, please share the results and any insights gained from this analysis.

The quality of English is good.

Author Response

Dear Reviewer,

Best Regards

Reviewer 2 Report

Need to improve

Author Response

Dear Reviewer,

Best Regards

Round 2

Reviewer 1 Report

Please check if there are any additional comments from other reviewers. Regards

The English language quality is satisfactory overall, but it would be beneficial to make some minor edits for improvement.